# MTHFR Gene Polymorphisms and Cardiovascular Risk Factors, Clinical-Imagistic Features and Outcome in Cerebral Venous Sinus Thrombosis

**DOI:** 10.3390/brainsci11010023

**Published:** 2020-12-27

**Authors:** Anca Elena Gogu, Dragos Catalin Jianu, Victor Dumitrascu, Horia Ples, Alina Zorina Stroe, Daniel Docu Axelerad, Any Docu Axelerad

**Affiliations:** 1Department of Neurology, University of Medicine and Pharmacy “Victor Babes”, 300041 Timisoara, Romania; agogu@yahoo.com (A.E.G.); dcjianu@yahoo.com (D.C.J.); 2Department of Pharmacology, University of Medicine and Pharmacy “Victor Babes”, 300041 Timisoara, Romania; dumitrascu.victor@umft.ro; 3Department of Neurosurgery, University of Medicine and Pharmacy “Victor Babes”, 300041 Timisoara, Romania; horia.ples@neuromed.ro; 4Department of Neurology, General Medicine Faculty, “Ovidius” University, 900470 Constanta, Romania; docuaxi@yahoo.com; 5Department of Sport, Faculty of Physical Education and Sport, “Ovidius” University, 900470 Constanta, Romania; docuaxy@yahoo.com

**Keywords:** cerebral venous sinus thrombosis, methylenetetrahydrofolate reductase gene, thrombophilia

## Abstract

Cerebral venous sinus thrombosis (CVST) as a severe neurological emergency, is represented by variable conditions in its clinic presentation, onset, risk factors, neuroimagistic features and outcome. The genetic polymorphism of the methylenetetrahydrofolate reductase *(MTHFR)* gene *C677T* and *A1298C* was associated with CVST. We aimed to characterize the prevalence of *MTHFR* gene polymorphisms associated with cardiovascular risk factors in the group of patients with CVST. Also, we studied additional causes associated with CVST including local infections, general infections, obstetric causes (pregnancy, puerperium) and head injury. This is a retrospective study including 114 patients which referred to our hospital between February 2012–February 2020. The protocol included demographic (age, sex), clinical, neuroimagistic features, paraclinic (genetic polymorphism of *MTHFR*, factor V *G1691A*—Leiden, prothrombin *G20210A*, PAI-1 *675 4G/5G*; Homocysteine level, the lipid profile, blood glucose and Glycohemoglobin HbA1c, high- sensitive C- reactive protein- hsCRP) data, as well as treatment and outcome. The mean age was 37.55 years with a female predominance (65.79%). In the first group of patients with inherited thrombophilia (60 cases; 52.63%) we found genetic mutation includes *MTHFR C677T* (38.59%) and *A1298C* (14.03%), factor V *G1691A*- Leiden (15.78%), prothrombin *G20210A* (2.63%), PAI-1 *675 4G/5G* (42.98%), and hyperhomocysteinemia (35.08%). At the second group with other etiology of CVST, except thrombophilia, we included 54 patients (47.36%). The most common sites of thrombosis were the superior sagittal sinus (52.63%). Headache was the most common symptom (91.22%) and seizures were the main clinical presentation (54.38%). The MTHFR polymorphism was significantly correlated with higher total cholesterol (TC) (*p =* 0.023), low- density lipoprotein cholesterol (LDL) (*p =* 0.008), homocysteine level (tHcy) (*p* < 0.001). Inside the first group with *MTHFR* polymorphism we have found a significant difference between the levels of homocysteine at the patients with *MTHFR C677T* versus *MTHFR A1298C* polymorphism (*p <* 0.001). The high-sensitive C-reactive protein (hsCRP) was increased in both groups of patients, but the level was much higher in the second group (*p =* 0.046). Mortality rate was of 2.63%. Demographic, clinical and neuroimagistic presentation of CVST in our study was similar with other studies on the matter, with a high frequency of thrombophilia causes. *MTHFR* gene polymorphisms (*C677T* and *A1298C*) are increased in prevalence in CVST. PAI-1 *675 4G/5G* gene mutation seems to be involved in CVST etiology. Plasma C-reactive protein level and hyperhomocysteinemia should be considered as a prognostic factor in CVST.

## 1. Introduction

Cerebral venous sinus thrombosis (CVST) represents an infrequent disorder responsible for less than 1% of all strokes [1] that can lead to significant morbidity and mortality, affecting young individuals. CVST, a severe neurological emergency is a variable condition in its clinical presentation, onset, risk factors, neuroimaging and outcome, making the diagnosis challenging [1,2]. CVST was considered an infectious disease for a long period but more recently its pathogenesis has been shown to be multifactorial with a key role for thrombophilia [3]. In the International Study on Cerebral Vein and Dural Sinus Thrombosis (ISCVT) genetic and acquired thrombophilia includes protein C, protein S and antithrombin deficiency, factor V Leiden mutation, prothrombin gene mutation as well as hyperhomocysteinemia [4]. Genetic polymorphism of methylenetetrahydrofolate reductase (*MTHFR*) gene *C677T* and *A1298C* was lately correlated with CVST. The aim of our study was to investigate the prevalence of *MTHFR* gene polymorphisms, factor V *G1691A*-Leiden mutation, prothrombin *G20210A* mutation, plasminogen- activator inhibitor *PAI-1 675 4G/5G* mutation and hyperhomocysteinemia correlated with cardiovascular risk factors in the group of patients with CVST. Hyperhomocysteinemia is considered an independent risk factor, potentially modifiable, for cardioembolic stroke but it is possible to exist the link with CVST. Cytosine (C) to thymine (T) substitution at position 677 in the *MTHFR* gene represents the most studied genetic variant, revealing the strongest correlation with elevated homocysteine (tHcy) values [4]. Apparently, the *MTHFR* gene could have a role in lipid metabolism too [5]. Hermans et al. [6] suggest that the *MTHFR* C677T polymorphism confers a higher risk for stroke to both homozygous and heterozygous T allele carriers. The impact of the *MTHFR* polymorphism on stroke may result from T allele-linked deleterious effects or C allele-linked protection; however, the authors consider that more studies are needed to support this hypothesis [6]. Also, we studied additional causes and risk factors correlated with CVST including local infections, general infections, obstetric causes (pregnancy, puerperium), head injury, direct injury after ethmoidal puncture for frontal sinusitis.

## 2. Materials and Methods

### 2.1. Patient Population

This is a retrospective study including 114 patients (75 females and 39 males) referred to the Neurology Department of Timisoara County Emergency Clinical Hospital between February 2012–February 2020 among 12,840 stroke admission, which represent 0.88%. Demographic (sex, age), clinical, computed tomography, magnetic resonance imaging (MRI), magnetic resonance venography (MRV), paraclinic (genetic polymorphism of *MTHFR*, factor V *G1691A*-Leiden, prothrombin *G20210A, PAI-1 675 4G/5G*, homocysteine level, the lipid profile, blood glucose and glycohemoglobin –HbA1c, high- sensitive C-reactive protein –hsCRP) data as well as other cardiovascular risk factors such as hypertension, systolic blood pressure, diastolic blood pressure, carotid atheromatosis detected with extracranial Doppler ultrasound and measured by intima- media thickness (IMT), were all considered.

We classified patient population into two groups. In the first group we included 60 patients with genetic thrombophilia, with *MTHFR* polymorphisms *C677T* and *A1298C*. These patients had also other genetic mutation like factor V *G1691A*- Leiden mutation (18 cases; 15.78%), prothrombin *G20210A* mutation (3 cases; 2.63%), *PAI-I 675 4G/5G* mutation (49 cases; 42.98%) without infectious causes. At the second group with other etiology of CVST, except thrombophilia, we included 54 patients. Local infections such as otitis, mastoiditis, maxilar sinusitis, dental abscess of the upper jaw and general infections like meningoencephalitis with HZV were all considered. We included into the second group patients with any condition that leads to a prothrombotic state such as pregnancy, puerperium, oral contraceptives use, complicated with septic conditions. In this second group with septic intracranial venous and sinus thrombosis we identified patients with direct head injury or after an ethmoidal puncture for frontal sinusitis. In the study were included patients previously diagnosed with comorbidities such as hypertension, carotid atheromatosis, type II diabetes mellitus (T2DM) and dyslipidemia. The exclusion criteria were: diagnosis of brain tumors and other oncologic pathologies and also psychiatric. The study was approved by Ethics Committee for Clinical Studies of the Timisoara County Emergency Clinical Hospital (Registration number: 4/29.01.2019, conformed to the Declaration of Helsinki). The written informed consent for study enrollment was obtained from every patient which was part of the study.

### 2.2. Clinical and Biochemical Evaluation

The clinical evaluation included a physical neurological examination, evaluation of neurological clinical condition and also the measurement of systolic and diastolic blood pressure. Blood pressure was determined according with the European Guidelines on Cardiovascular Disease Prevention in Clinical Practice [7]. An electrocardiogram was performed at every patient on admission for excluded acute heart failure.

Depending on the period between the onset of the clinical symptoms and the time of diagnosis, the disease stages were designed as acute (< 48 h), subacute (between 48 h and 30 days) and chronic (more than 30 days), as proposed by the International Study on Cerebral Venous and Dural Sinuses Thrombosis [8].

The paraclinical blood related investigations were performed for each patient on admission and the results were provided by Medical Laboratory of Timisoara County Emergencies Clinical Hospital. The lipidogram, respectively total cholesterol (TC), low- density lipoprotein cholesterol (LDL), high- density lipoprotein cholesterol (HDL), triglycerides (TGL) was determined using photometric methods. Glycohemoglobin (HbA1c) was determined with immunoturbidimetric assay and the results were interpreted according to American Diabetes Association (ADA) recommendations: values between 4.8–5.6% as normal, 5.7–6.4% values representing a high risk for developing T2DM and values higher than 6.5% as T2DM [9,10]. High- sensitive C- reactive protein (hsCRP) was determined using a quantitative immunoturbidimetric assay, with a linearity method between 0.01–160 mg/L. Reference interval ≤ 5mg/L was used for interpretation [10]. Because elevated plasma homocysteine levels are considered a special risk factor for cardioembolic stroke in numerous studies [11,12,13], we included in our study this variable, evaluated a possible link with CVST. A normal level of homocysteine in the blood is less than 15 µmol /L of the blood. Higher levels of homocysteine are split into three main categories: moderate 15–30 µmol/L; intermediate 30–100µmol/L; severe- greater than 100 µmol/L [14].

### 2.3. Genetic Testing

We made genetic testing of *MTHFR* (*C677T* and *A1298C*) polymorphisms, factor V *G1691A*- Leiden mutation, prothrombin *G20210A* gene mutation, plasminogen activator- inhibitor *PAI-1 675 4G/5G* gene mutation. The intervention was performed according to the Institution’s protocols, thereby, an amount of 3–6 mL of peripheral blood was collected from each patient in EDTA vacutainers. Genetic testing was performed at Bioclinica Laboratories (Timisoara, Romania). Genomic DNA was extracted using QIAamp DNA mini blood kits (Qiagen, Hilden, Germany). PCR amplification was made using LightCycler 480 Instrument II Platform according to the manufacturer’s (Roche Diagnostics GmbH, Mannheim, Germany) manual instructions.

### 2.4. Neuroimaging

A non-contrast computer tomography (CT) was performed at every patient on admission to the emergency room. The speed and accessibility with which this test can be obtained make it the first test that we used in the clinical evaluation of any patient who presents with atypical headache, focal neurological deficit, seizures, altered mental status or coma. A direct sign of cerebral venous thrombosis is the cord sign which is a curvilinear hyperdensity within a cortical vein in the presence of thrombosis that can be seen for up two weeks following thrombus formation [15,16]. Other direct signs include hyperdensity with a triangular shape in the superior sagittal sinus, also known as the dense triangle sign. Intraparenchimal hemorrhages, subarachnoid hemorrhage or infarcts may be seen on non- contrast cerebral CT [15,16], but in some cases the CT scan was normal.

The imagistic evaluation by magnetic resonance imaging (MRI) and magnetic resonance venography (MRV) was performed for every patient between the first 48 h of admission. MRI sequences were performed with a 1.5- T MR unit (Siemens Medical Systems, Erlangen, Germany), including T1- and T2-weighted sequences, fluid- attenuated inversion recovery (FLAIR) imaging sequence, diffusion- weighted imaging (DWI) sequences with apparent diffusion coefficient (ADC) calculations, and susceptibility- weighted imaging (SWI). Neuroimaging MRIs were performed and interpreted by radiologists at the Timisoara County Emergencies Clinical Hospital. MR-angiography (three- dimensional time-of-flight 3D-TOF sequences) and MR-venography (2D-TOF) were obtained during the same imaging session [17]. Considered to be the gold standard in diagnosing cerebral venous thrombosis as it has a higher sensitivity than computer tomography [17]. MRI is superior to CT when evaluating for parenchymal edema as a result of cerebral venous thrombosis. MRI findings are dependent on age of the thrombus, as signal intensities change depending thrombus age. Thus, MRI interpretation requires a detailed understanding of the evolutionary changes that are seen radiographically. An acutely formed thrombus (0–7 days) is harder to detect, but by week two abnormalities are easier to detect, with both T1 and T2- weighted images showing a hyperdense signal [17]. The combination of abnormal signal in a venous sinus combined with the absence of flow on MRV confirms the diagnosis of CVST. The venogram in minimum two plans is useful (axial, coronal and/or sagittal). The difference of the venous and sinus thrombosis from anatomical versions and signal loss by flow artifacts flux is difficult and may give positive results [18]. The false negative in the case of the dural sinuses thrombosis may be given by a subacute thrombus with strong T1 hypersignal. At MR venography the superior longitudinal sinus and the right sinus are given with an accuracy of 100%; the transverse sinus with an accuracy of 95% and the inferior longitudinal sinus with 45% [18]. The parenchymatous lesions occurs with hypersignal in T2 for localized or ischemic edema. In the hemorrhagic lesions T1 and T2 hypersignals occur often surrounded by a hypersignal black ring [18].

### 2.5. Ultrasound Evaluation of Carotid Arteries

Extracranial Doppler ultrasound for carotid arteries was performed in all patients included in the study using an Acuson SC2000 Prime ultrasound system (Siemens Healthcare GmbH, Erlangen, Germany). In screening to investigate the presence of carotid atheromatosis, we have identified intima- media thickness (IMT) measurements, as presenting values of <0.9 mm considered as normal. The presence/absence of carotid plaques and carotid stenosis were also considered.

### 2.6. Statistical Analyses

Statistical calculations were performed using the Microsoft Excel (version 2016, Microsoft Corp., Redmond, WA, USA). The presented dataset was hypothesis- driven research.

A Chi square test was the major method used to compare different variables of the diagnosis and numerical information. The Chi Square Test is a test that involves the use of parameters to test the statistical significance of the observations under study. Because is a nonparametric test of the statistical significance of a relation between two nominal or ordinal variables it was used to analyze grosser data than do parametric tests such as t tests and analyses of variance. Considering this, the chi-square test reports whether groups are significantly different in some measured attribute or behavior.

Fisher’s exact test was used to determine nonrandom associations between two categorical variables.

Odds ratio (OR) is a measure of the strength of association with an exposure and an outcome being used for the analysis of variables. The 95% confidence interval (CI) was considered to estimate the precision of OR. A large CI indicates a low level of precision of the OR, whereas a small CI indicates a higher precision of the OR.

Moreover, in this article P stands for probability and measures how likely it is that any observed difference between groups is due to chance. Considering this, *p*-value < 0.05 was used for significant differences. Statistical software gave the exact *p*-value and allowed appreciation of the range of values.

Outcome of the multiple diagnoses with were compared using the Wilcoxon-Mann-Whitney two-sample rank-sum test. The difference was quantified using the Hodges-Lehmann (HL) estimator (*p*-value < 0.001). This estimator (HL∆) was considered the median of the all possible differences in outcomes.

The Mann–Whitney U-test is a distribution-free test used to determine whether two independent groups have been drawn from the same population. Mann-Whitney U test is used to test for differences in medians. For the current scope, the test statistic U is calculated from comparing each pair of values, one from each group, scoring these pairs 1 or 0 depending on whether the first group observation is higher or lower than that from the second group and summing the resulting scores over all pairs. When the median is 0, the *p <* 0.0001. The calculated test statistic is then referred to the appropriate table of critical values to decide whether the null hypothesis of no difference in the location of the two data sets can be rejected.

## 3. Results

### 3.1. Patient Population

#### 3.1.1. Demographic Characteristics

A total 114 patients with CVST were recruited for this study. There were 75 females and 39 males (65.78% versus 34.21%). The mean age of the patients was 37.55 years (standard deviation SD = 11.86; range = 18–70 years). In the age groups by decades, the highest frequency was seen in the third decade (49 cases, 42.98%).

Patients’ characteristics including demographic, clinical, neuroradiological and biochemical data are summarized in Table 1.

#### 3.1.2. Etiology of CVST

Many risk factors contribute to the development of CVST. We identified multiple risk factors in more than 97% of patients and only one risk factor at 3% of patients with CVST. Inherited thrombophilia includes factor V *G1691A*-Leiden mutation (18 cases; 15.78%), prothrombin *G20210A* gene mutation (three cases; 2.63%), *MTHFR* gene polymorphism *C677T* (44 cases; 38.59%) and *A1298C* (16 cases; 14.03%), *PAI-1 675 4G/5G* gene mutation (49 cases; 42.98%), as well as hyperhomocysteinemia (40 cases; 35.08%) was found at 60 patients (52.63%) from the first group.

At the second group with other etiology of CVST, except thrombophilia, we included 54 patients (47.36%). Local infections such as otitis, mastoiditis, maxillary sinusitis, dental abscesses of the upper jaw (27 cases; 23.68%) and meningoencephalitis with HZV (one case; 0.87%) has been identified. CVST was found in any condition that leads to a prothrombotic state, including pregnancy (four cases; 3.5%), puerperium (four cases; 3.5%) or those on oral contraceptives use (14 cases; 12.28%). CVST may also be seen in the patients with head injury (four cases; 3.5%) and even after an ethmoidal puncture for frontal sinusitis (one case; 0.87%). We mentioned that the patients from the second group do not presented any genetic mutations (*MTHFR*, factor V-Leiden, prothrombin and *PAI-1*) but 24 patients (21.05%) had hyperhomocysteinemia.

#### 3.1.3. Clinical Findings

The mode of onset was acute (<48 hours) in 25 patients (21.92%), subacute (48 hours-30 days) in 80 patients (70.17%) and chronic (more than 30 days) in nine patients (7.89%).

The symptomatology varied according to the disease etiology and the location of sinus thrombosis. CVST can present with variable sign and symptoms that include headache (104 cases; 91.22%), benign intracranial hypertension (74 cases; 64.91%), focal neurological deficit like hemiparesis, ataxic tetraparesis (40 cases; 35.08%), focal and generalized seizures (62 cases; 54.38%), unexplained altered sensorium at onset (34 cases; 29.82%), cranial nerve palsies (30 cases; 26.31%), nausea and vomiting (94 cases; 82.45%), papilledema (24 cases; 21.08%), coma at onset (three cases; 2.63%).

Patients with a history of previously documented comorbidities such as hypertension, carotid atheromatosis, type II diabetes mellitus (T2DM) and dyslipidemia were included in the study. We found 24 patients (21.05%) with hypertension. The mean value of the systolic blood pressure was 126.53 mmHg (SD, 23.39) and the mean value of diastolic blood pressure was 77.85 mmHg (SD, 13.18) at the entire study group (*n* = 114).

### 3.2. Neuroradiological Findings

A non-contrast computed tomography (CT) scan was performed at every patient on admission to the emergency room. The CT scan was normal in 56 patients (49.12%). A direct sign of cerebral venous thrombosis, the corn sign, was found in 32 patients (28.07%) and intraparenchimal hemorrhages in 28 patients (24.56%).

All patients underwent imagistic evaluation by magnetic resonance imaging (MRI) and magnetic resonance venography (MRV) during the first 48 h after admission. The most common sites of thrombosis were the superior sagittal sinus (60 cases; 52.63%), the lateral sinus (37 cases; 32.45%) and cavernous sinus (17 cases; 14.91%). The sigmoid sinus (four cases; 3.5%) and petrosal sinus (two cases; 1.75%) were also involved. More than one sinus was involved in 75 cases (65.78%). Parenchymal lesions were observed in 68 patients (59.64%) and most of them had hemorrhagic infarcts localized at the fronto-parietal lobe or at the temporal lobe, related with the topography of the cerebral sinus thrombosis. Cerebral edema was observed at 74 cases (64.91%) and subarachnoid hemorrhage at 11 patients (9.64%).

We describe next the most special cases, having a various clinical symptomatology, correlated with neuroradiological findings.

The first case, was a female, 26 years old, with an etiology of left otomastoiditis, hospitalized in superficial coma, with meningeal symptoms and HIC, psychomotor agitation and three generalized seizures. The Grisinger sign was present at local examination of mastoiditis. After a period of six months, at the reevaluation, the same patient did not present any residual symptomatology related to thrombophlebitis. The imagistic features can be seen in Figure 1 and Figure 2.

The second case was a female, 31 years old, with an etiology of thrombophlebitis accentuated by the presence of the *MTHFR* C677T gene polymorphism- homozygous phenotype. The patient was hospitalized with superficial coma, generalized seizures and left hemiparesis. In the course of evolution, subsequent to the cerebral post thrombotic syndrome and edematous encephalopathy, the patient presented symptomatology characterized by generalized seizures under anticonvulsivant treatment and also psychical disturbances (irritability, aggressiveness). The patient was clinically and neuroimagistic assessed after six month and six years from the onset of the disease. The image features can be seen in Figure 3, Figure 4 and Figure 5.

Extracranial Doppler ultrasound was performed in all patients included in the study in order to detect carotid atheromathosis with intima-media thickness measurements for the evaluation of cardiovascular risk factors (mean IMT = 0.88 mm; SD, 0.15).

### 3.3. Biochemical Findings

Because the *MTHFR* gene seems to be involved in lipid metabolism, we considered it important in our study to detect the total cholesterol (TC; mean value = 195.48 mg/dL; SD, 37.36), low- density lipoprotein cholesterol (LDL; mean value = 105.34 mg/dL; SD, 28.44), high- density lipoprotein cholesterol (HDL; mean value = 51mg/dL; SD, 12.64) and triglycerides (TGL; mean value = 140.75 mg/dL; SD, 77.29).

Blood glucose and glycohemoglobin (HbA1c) was determined at all patients and the results were interpreted according to American Diabetes Association (ADA) recommendations. We found at blood glucose the mean value 95.5 mg/dL, the standard deviation (SD), 29.41 and at the glycohemoglobin (HbA1c) the mean value was 5.32, the standard deviation (SD), 1.13.

High- sensitive C-reactive protein (hsCRP) was determined at 114 patients and the mean value was 10.58 mg/L and standard deviation (SD), 11.75.

The elevated plasma homocysteine levels (tHcy) are considered a special risk factor for cardioembolic stroke in many studies and for this reason we included this variable in our study, evaluated a possible link with CVST. At our patients (*n* = 114) the mean value was 24.10 µmol/L and standard deviation (SD), 27.25.

We studied two groups of patients. In the first group we included 60 patients with inherited thrombophilia; all of them had *MTHFR* gene polymorphism (*C677T* and/or *A1298C*) but some of them had other polymorphism besides *MTHFR*. We found 44 cases (38.59%) with *MTHFR* gene polymorphism *C677T* and 16 cases (14.03%) with *MTHFR* gene polymorphism *A1298C*; 52 cases (45.61%) presented the homozygous genotype. In the group with *MTHFR* gene polymorphism we also included patients with factor V *G1691A*- Leiden mutation (18 cases, 15.78%), prothrombin *G20210A* gene mutation (3 cases, 2.63%), *PAI-1 675 4G/5G* gene mutation (49 cases, 42.98%) as well as hyperhomocysteinemia (40 cases, 35.08%) without infectious causes of CVST.

At the second group with other etiology of CVST, except thrombophilia, we included 54 patients (47.36%). We mention that the patients with septic CVST do not present any genetic mutations but 24 patients (21.05%) had hyperhomocysteinemia.

By comparing patients with inherited thrombophilia (*n* = 60) with *MTHFR* polymorphism versus patients with other etiology of CVST (*n* = 54) significantly increased values for TC (201.48 ± 38.54 mg/dL vs. 188.81± 35.16 mg/dL; *p =* 0.023), LDLc (115.35 ± 27.47 mg/dL vs. 94.25± 25.44 mg/dL; *p =* 0.008), tHcy (34.79 ± 34.02 µmol/L vs. 12.23 ± 4.72 µmol/L; *p* < 0.001), hsCRP (7.12 ± 2.88 mg/L vs. 14.44 ± 16.02 mg/L; *p =* 0.046) were found in the first group. There were no significant differences between groups regarding age (*p =* 0.221), male (*p =* 0.167), female (*p =* 0.356), HTN (*p =* 0.172), SBP (*p =* 0.091), DBP (*p =* 0.132), SSS/SSS+LS (*p =* 0.158), LS/LS+PS/LS+SS (0.496), Cav.S (*p =* 0.585), HDL (*p =* 0.287), TGL (*p =* 0.071), blood glucose (*p =* 0.061), HbA1c (*p =* 0.699), IMT (*p =* 0.859) (Table 1).

The comparison between patients with *MTHFR* C677T polymorphism (*n* = 44) and patients with *MTHFR A1298C* polymorphism (*n* = 16) revealed a significant differences between groups regarding gender, male (16± 36.37% vs. 6 ± 37.50%; *p <* 0.001), female (28 ± 63.63% vs. 10 ± 62.50%; *p <* 0.001), HTN (10 ± 22.72% vs. 4 ± 25%; *p =* 0.002), SSS/ SSS+ LS (25 ± 56.81% vs. 7 ± 43.75%; *p <* 0.001), LS/ LS+ SP/LS+ SS (15 ± 34.09% vs. 5 ± 31.25%; *p <* 0.001), factor V- Leiden (16 ± 36.36% vs. 2 ± 12.50%; *p <* 0.001), PAI-1 (38 ± 86.36% vs. 11 ± 68.75%; *p <* 0.001), tHcy (45.05 ± 34.13 µmol/L vs. 17.19 ± 34.31µmol/L; *p <* 0.001). There were no significant difference between groups regarding age (*p =* 0.548), SBP (*p =* 0.264), DBP (*p =* 0.326), Cav. S (*p =* 0.844), factor II (*p =* 0.082), TC (*p =* 0.108), LDLc (*p =* 0.292), HDLc (0.385), TGL (0.149), blood glucose (*p =* 0.316), HbA1c (*p =* 0.817), hsCRP (*p =* 0.745), IMT (*p =* 0. 923) (Table 2).

Using the dates from Table 1 and Table 2, we compared the group of patients with *MTHFR C677T* gene polymorphism (*n* = 44) with the group of patients with other CVST etiology (*n* = 54). We obtained statistically significant increased values for TC (198.45 ± 38.36 mg/dL vs.188.81 ± 35.16 mg/dL; *p* < 0.005), LDLc (115.2 ± 26.76 mg/dL vs. 94.25 ± 25.44 mg/dL; *p <* 0.005), tHcy (45.05 ± 34.13 μmol/L vs. 12.23 ± 4.72 μmol/L; *p <* 0.005), IMT (0.92 ± 0.18 mm vs. 0.83 ± 0.1 mm; *p* < 0.005). Furthermore, in the group of patients with *MTHFR C677T* gene polymorphism, the following was noted: the number of women and the measured values of cavernous sinus were statistically significant lower compared with the group of patients with other etiology of CVST (28 vs. 37; 4 vs.9; *p <* 0.005).

We obtained the following conclusions, by comparing the values obtained by the group patients with *MTHFR A1298C* gene polymorphism (*n* = 16) with the values obtained by the group of patients with other etiology of CVST (*n* = 54): the results showed statistically significant increased values for IMT (0.94 ± 0.18 mm vs. 0.83 ± 0.01 mm; *p <* 0.005). Also, the results revealed statistically significant decreased values between the two groups regarding the following variables: male (6 vs. 17; *p <* 0.005), female (10 vs. 37; *p <* 0.005), SSS/SSS+LS (7 vs. 27: *p <* 0.005), LS/LS+SP/LS+SS (five vs. 18; *p <* 0.005).

### 3.4. Treatment

Regarding treatment methods, all patients (*n* = 114) were treated with unfractionated heparin, on average for 14 days, followed by an oral anticoagulation, with the international normalized ratio maintained between 2 and 3. The oral anticoagulation decision and duration of this treatment was individualized. Etiological treatment was performed for septic CVST, wide range cephalosporins (54 cases, 47.36%). Symptomatic treatment was administered to all patients with the anticonvulsivant therapy since hospitalization and it has been continued for those with seizures episodes even in the present (levetiracetam, valproic acid, diazepam, propofol). Therapy of depletion with mannitol was performed for the patients with intracranial hypertension and in the presence of papillary edema (74 cases, 64.91%) and steroids in 24 cases (21.08%).

### 3.5. Evolution

The number of hospitalization days ranged between 5–35, with an average of 16 days. Three deaths (2.63%) occurred in the acute phase in cases of patients with septic CVST and coma at onset. Generally, the evolution was favorable. Six-month follow-up was available for 85 patients (74.56%). Of these, sixty patients (70.58%) displayed complete recovery, fifteen patients (17.64%) had partial recovery and seven cases (8.23%) were dependent. As residual symptoms, we included focal motor deficits, epilepsy and cerebral post-thrombotic syndrome, accompanied by psychic disorders. These last cases showed superior sagittal sinus thrombosis, a possible explanation could be the involved of the frontal lobe.

## 4. Discussion

Cerebral venous sinus thrombosis is a rare disorder responsible for less than 1% from the total of stroke, with a higher frequency among young adults, females during pregnancy, the puerperium or who take oral contraceptives, and patients with thrombophilia [17,18,19]. CVST can present with variable sign and symptoms that include a headache, benign intracranial hypertension, focal neurological deficits, seizures, unexplained altered sensorium, which are often misleading. Thus, having a high index of suspicion for this disease is crucial to ensure timely diagnosis and treatment [20,21,22]. CVST is an uncommon cause of stroke accounting 0.88% of all our hospitalized patients with stroke (114 cases among 12,840 cases).

In our study, we found a young female predominance (65.79% vs. 34.21%) and the highest frequency of occurrence in the third decade (49 cases, 42.98%). The mean age of the patients was 37.55 years (SD = 11.86, range 18–70 years).

The onset was subacute in most of cases (80 cases, 70.17%), depend on various factors, especially on the site, extent and rate of progression of thrombosis. The symptomatology was complex, in relation with the etiology of disease and the location of sinus thrombosis. Headache was the most common symptom (104 cases, 91.22%) and benign intracranial hypertension was the main clinical presentation (74 cases, 64.91%), followed by focal or generalized seizures (62 cases, 54.38%) and focal neurological deficit (40 cases, 35.08%).

The most common sites of CVST in our study were the superior sagittal sinus (60 cases, 52.63%), the lateral sinus (37cases, 32.45%) and cavernous sinus (17 cases, 14.91%). More than one sinus was involved in 75 cases (65.78%). There were no significant differences between group with inherited thrombophilia and group with septic CVST regarding location of thrombosis: superior sagittal sinus 33 cases (55%) in the first group versus 27 cases (50%) in the second group (*p =* 0.158); lateral sinus 19 cases (31.66) vs. 18 cases (33.33%) (*p =* 0.496); cavernous sinus eight cases (13.33%) versus nine cases (16.66%) (*p =* 0.585). The intracranial venous anatomy is complex, and anatomic variations of important intracranial venous structures are common. Cerebral lesions and clinical syndromes resulting from CVST occur in patterns directed related to the venous anatomy and infarctions usually cross a particular arterial distribution, often with hemorrhagic transformation [18]. The parenchymatous lesions are represented by venous infarctions, most often hemorrhagic, involving the cortex and the white matter close to the thrombosed vein or sinus [23].

Many risk factors contribute to development of CVST. We identified multiple risk factors in more than 97% of patients and only one risk factor at 3% of patients with CVST. We studied two groups of patients. In the first group we included 60 patients (52.63%) with inherited thrombophilia; all of them had the *MTHFR* gene polymorphism (*C677T* and/or *A1298C*) but some of them had other polymorphisms besides *MTHFR*. We found 44 cases (38.59%) with *MTHFR* gene polymorphism C677T and 16 cases (14.03%) with *MTHFR* gene polymorphism *A1298C*; 52 cases (45.61%) presented the homozygous genotype. In this first group we also included patients with factor V *G1691A*- Leiden mutation (18 cases, 15.78%), prothrombin *G20210A* gene mutation (three cases, 2.63%), *PAI-1 675 4G/5G* gene mutation (49 cases, 42.98%) as well as patients with hyperhomocysteinemia (40 cases, 35.08%) without infectious causes of CVST. The genetic variants of the *MTHFR* gene, respectively C677T and *A1298C* polymorphisms, have been study by numerous authors over time. Although these genetic polymorphisms have proven to be inherited and associated with several clinical conditions, for example cardioembolic stroke [24].

Our study aimed to look for the associations between genetic variants of the *MTHFR* gene in CVST, localization and clinical severity of disease and also additional cardiovascular risk factors involved in CVST. In the second group with other etiology of CVST, except thrombophilia, we included 54 patients (47.36%). Local and general infection has been identified at 28 patients (24.56%). CVST was found in any condition that leads to a prothrombotic state, including pregnancy (four cases, 3.5%), puerperium (four cases, 3.5%) or those on oral contraceptives use (14 cases, 12.28%). CVST may also be seen in the patients with head injury (four cases, 3.5%) and even after an ethmoidal puncture for frontal sinusitis (one case, 0.87%). Septic factors involved in etiology of CVST are in a large percentage in our study. Puerperium and pregnancy, related to a transient prothrombotic state, were not a major risk factor identified in our patients. In contrast to previous Western reports and similarly to some Asian studies, oral contraceptives were not a major risk factor for CVST in our patients (12.28%) [8,25,26]. A possible explanation would be that the patients with oral contraceptives use did not have any genetic mutation (*MTHFR*, factor V-Leiden, factor II, PAI-1).

By comparing patients with inherited thrombophilia (*n* = 60) with *MTHFR* polymorphism versus patients with other etiology of CVST (*n* = 54) significantly increased values for TC (*p =* 0.023), LDLc (*p =* 0.008), tHcy (*p <* 0.001), hsCRP (*p =* 0.046) were found in the first group. The *MTHFR* gene seems to be involved in lipid metabolism. Numerous studies regarding inflammation in stroke have been conducted so far but Chen et al. managed to find the association between neuroinflammation due to overrated microglial activation, augmented by elevated plasma level of tHcy in a rat model with ischemic stroke [26]. Hyperhomocysteinemia is considered an independent risk factor, conceivably adjustable, in relation with cardioembolic stroke but with the possibility of the presence of a connection with CVST. The hsCRP was increased in patients with *MTHFR* polymorphism (mean value = 7.12 mg/L) but at the patients with septic CVST the level was much higher (mean value = 14.44 mg/L) (*p =* 0.046).

The variables’ comparison between patients with *MTHFR C677T* polymorphism (*n* = 44) and patients with *MTHFR A1298C* polymorphism (*n* = 16) revealed a significant differences between groups regarding gender, male (*p <* 0.001), female (*p <* 0.001), HTN (*p =* 0.002), SSS/SSS+LS (*p <* 0.001), LS/LS+SP/LS+SS (*p <* 0.001), factor V-Leiden (*p <* 0.001), *PAI-1* (*p <* 0.001), tHcy (*p <* 0.001). We also analyzed the connection between the presence of the *C677T* polymorphism and CVST localization on the premise that the affected cerebral area is important for the clinical severity. Superior sagittal sinus is involved more frequent at patients with *MTHFR* C677T gene polymorphism. There was no significant difference between group with *C677T* and group with *A1298C* regarding TC, LDLc, HDLc, TGL, blood glucose, HbA1c, hsCRP and IMT.

By comparing the group of patients with *MTHFR C677T* gene polymorphism (*n* = 44) with the group patients with other etiology of CVST (*n* = 54) we obtained statistically significant increased values for TC, LDLc, tHcy, IMT.

We obtained the following conclusions, by comparing the values obtained by the group patients with *MTHFR A1298C* gene polymorphism (*n* = 16) with the values obtained by the group of patients with other etiology of CVST (*n* = 54): the results showed statistically significant increased values for IMT.

All our patients were treated with unfractionated heparin. Mortality rate was of 2.63%. Three deaths occurred in the acute phase at patient with septic CVST and coma at onset. The monitoring of patients’ evolution was possible for 85 patients and outcome was beneficial for 60 patients which presented complete recovery.

The limitation of our study was its retrospective nature with the recruitment of all patients from a single department. The relatively small number of patients and the high rate of patient’s evolution monitoring loss are other limitations of this study. In addition, some pro-thrombotic conditions such as protein C, protein S, antithrombin deficiency have not been investigated in our patients.

Our conclusions of the study are on the same line with the conclusions of other studies from the literature, performed in various parts of the globe, as follows: the higher levels than normal in plasma Hcy levels translates into hyperhomocysteinemia which represents a risk factor the pathophysiology of cardiovascular and cerebrovascular diseases and also in the recovery of the injuries located in the cerebral area as revealed in studies from literature [27,28,29]. In a study by Tripathi et al., following a North Indian population, *MTHFR C677T* polymorphism and higher plasma Hcy levels have also been related with coronary artery diseases [30]. *MTHFR* polymorphism referring to *C677T* and *A1298C*; has been implicated in hypercysteinemia revealing an increased occurrence of ischemic and hemorrhagic stroke in the Chinese population [31,32]. The data from the study of Kumar et al. has revealed the existence of a correlation between higher Hcy levels and *MTHFR C677T* polymorphism in subarachnoid hemorrhage patients [33]. Studies from literature [34,35,36,37] have emphasiezed the role of *MTHFR C677T* polymorphism in ischemic stroke. *MTHFR C677T* polymorphism conducts to substitution of alanine to valine at position 222 which encodes for a thermolabile enzyme with reduced activity. This decrease in enzyme activity could be accountable for higher Hcy levels. In the study by Kumar et al., *MTHFR A1298C* polymorphism has also been found to be accountable for the elevated Hcy levels in the Indian population [38].

## 5. Conclusions

Demographic, clinical and radiologic presentation of CVST in our study was similar with the conclusions provided by other studies on the same matter, with an increased frequency of causes which implied the presence of thrombophilia. *MTHFR* gene polymorphisms (*C677T* and *A1298C*) have a high prevalence in CVST. The *MTHFR* polymorphism was correlated with higher low- density lipoprotein cholesterol (LDLc), high-sensitive C- reactive protein (hsCRP), homocysteine level (tHcy). A correlation between *MTHFR* polymorphism and CVST severity was emphasized. *PAI-1 675 4G/5G* gene mutation seems to be involved in CVST etiology. In CVST caused by local infections or oral contraceptive drugs we should be considered increased hsCRP and hyperhomocysteinemia as a prognostic factor. Patients with septic CVST and altered consciousness from the beginning have bad prognosis.

## Figures and Tables

**Figure 1 brainsci-11-00023-f001:**
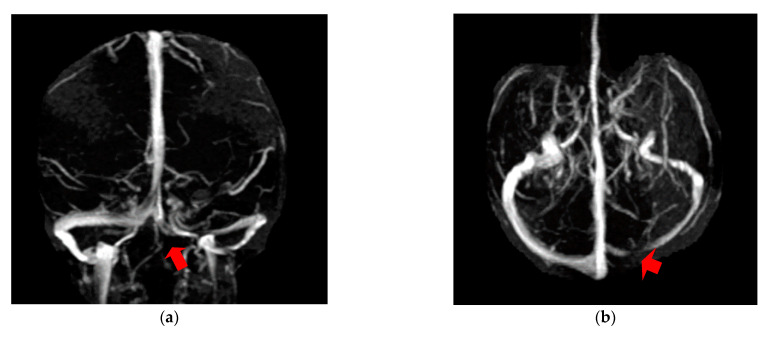
MR-venography 2D-TOF (2D- time of flight) in the coronal (**a**) and axial (**b**) plane noting the absence of the flow in the left lateral sinus (thrombosis of the left LS at the onset).

**Figure 2 brainsci-11-00023-f002:**
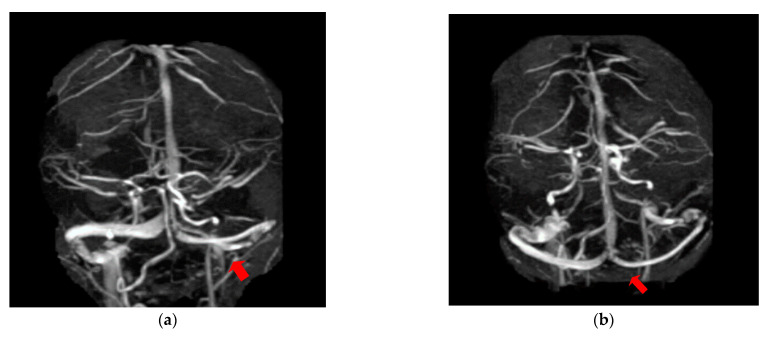
MR- venography 2D-TOF (2D- time of flight) in the coronal (**a**) and axial (**b**) plane reveals permeabilization of the left lateral sinus after six months from the onset.

**Figure 3 brainsci-11-00023-f003:**
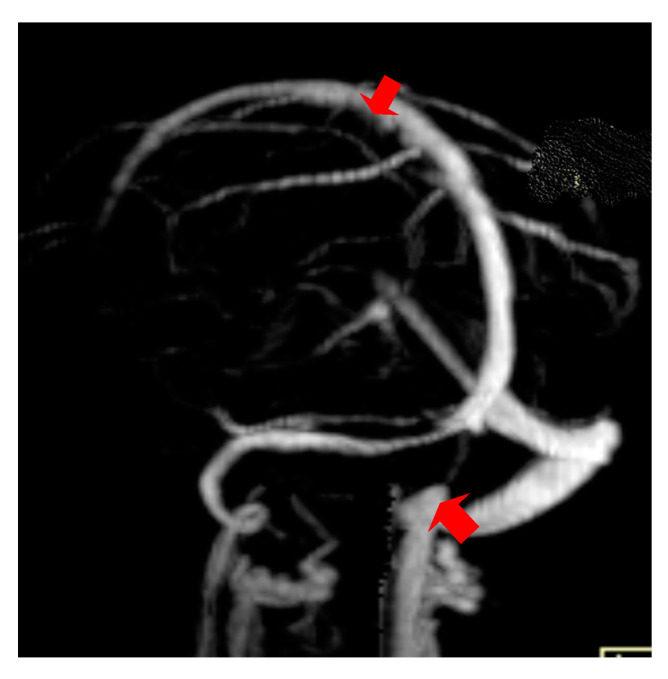
MR- venography Maximum Intensity Projection (MIP) range reveals thrombosis of the superior sagittal sinus and the right lateral sinus at the onset.

**Figure 4 brainsci-11-00023-f004:**
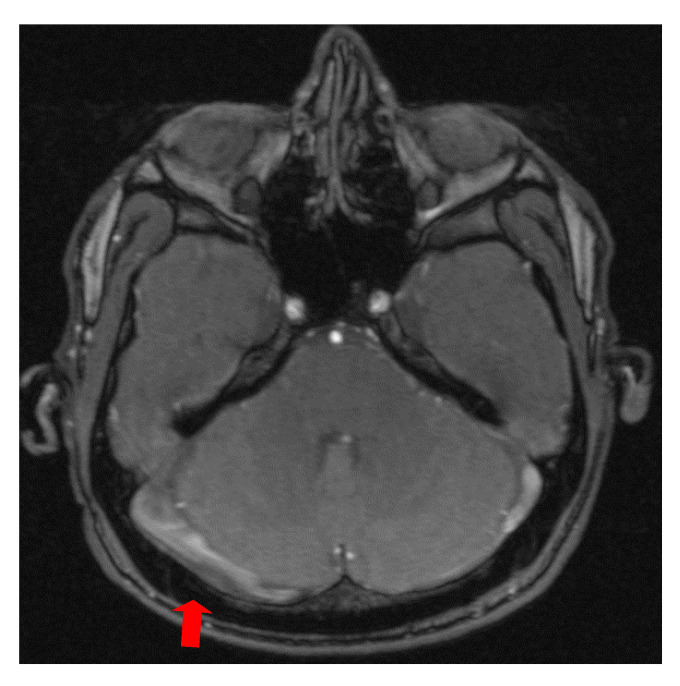
MR- venography 3D- TOF (time of flight) reveals subarachnoid hemorrhage which accompanies superior sagittal sinus and right lateral sinus thrombophlebitis.

**Figure 5 brainsci-11-00023-f005:**
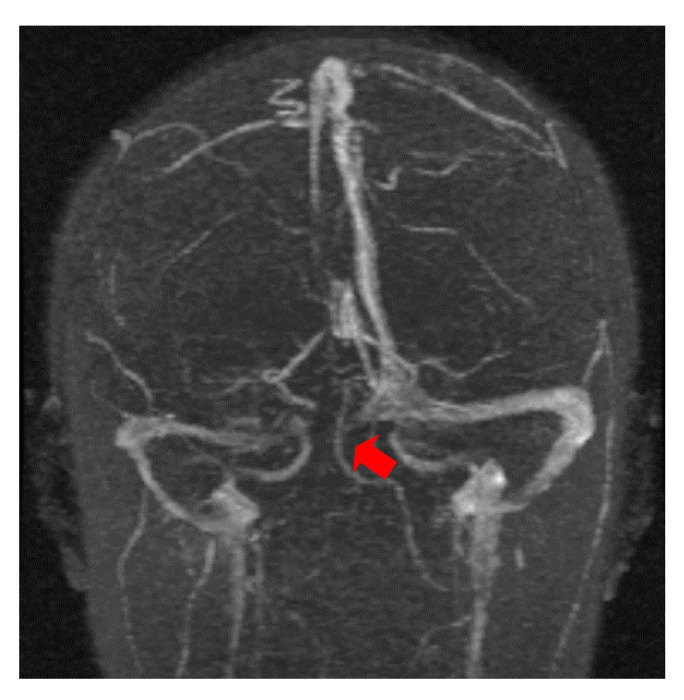
MR- venography 2D- TOF (time of flight) in the coronal plane reveals the permeabilization of the superior sagittal sinus but in the right lateral sinus persist the absence of the flow after six years from the onset of the disease.

**Table 1 brainsci-11-00023-t001:** Variable comparisons for *MTHFR* (*C677T* and *A1298C*) gene polymorphism (*n* = 60) and other etiology of CVST (*n* = 54) using the Mann- Whitney U Test and Chi Square Test.

Variable	Patients Total, *n* = 114	With *MTHFR* Polymorphism (*n* = 60)		With Other Etiology (*n* = 54)		*p*
	Mean ± Std Deviation	Mean ± Std Deviation (Median)	Mean Rank	Mean ± Std Deviation (Median)	Mean Rank	
Age, years	37.55 ± 11.86	35.1 ± 8.48	37.73	40.27 ± 14.33	37.63	0.221
Male, n (%)	39 (34.21%)	22 (36.66%)	21.44	17 (31.48%)	17.54	0.167
Female, n (%)	75 (65.79%)	38 (63.34%)	39.07	37 (68.51%)	35.90	0.356
HTN ^1^, n (%)	24 (21.05%)	14 (23.33%)	13.42	10 (18.55%)	10.56	0.172
SPB ^2^, mmHg	126.53 ± 23.39	128 ± 22.45	131.70	124.9 ± 24.50	121.13	0.091
DBP ^3^, mmHg	77.85 ± 13.18	80.91 ± 13.6	82.04	74.44 ±11.92	73.26	0.132
SSS ^4^/SSS + LS	60 (52.63%)	33 (55%)	32.56	27 (50%)	27.42	0.158
LS ^5^/LS+SP ^6^/LS+SS ^7^, n (%)	37 (32.45%)	19 (31.66%)	19.40	18 (33.33%)	17.58	0.496
Cav ^8^. S, n (%)	17 (14.91%)	8 (13.33%)	8.55	9 (16.66%)	8.44	0.585
Factor V ^9^, n (%)	18 (15.78%)	18 (30%)	13.81	NA	NA	NA
Factor II ^10^, n (%)	3 (2.63%)	3 (5%)	2.30	NA	NA	NA
PAI-1 ^11^, n (%)	49 (42.98%)	49 (81.66%)	37.61	NA	NA	NA
tHcy ^12^, µmol/L	24.1 ± 27.25	34.79 ± 34.02	29.98	12.23 ± 4.72	17.02	*p* < 0.001 *
TC ^13^, mg/dL	195.48 ± 37.36	201.48 ± 38.54	205.22	188.81 ± 35.16	184.96	0.023 *
LDLc ^14^, mg/dL	105.34 ± 28.44	115.35 ± 27.47	113.78	94.25 ± 25.44	95.76	0.008 *
HDLc ^15^, mg/dL	51 ± 12.64	51.35 ± 12.77	52.97	50.61 ± 12.6	48.96	0.287
TGL ^16^, mg/dL	140.75 ± 77.29	138.56 ± 75.97	144.70	143.18 ± 79.37	136.96	0.071
Blood glucose, mg/dL	95.5 ± 29.41	100.8 ± 39.33	101.37	89.61 ± 7.41	88.98	0.061
HbA1c ^17^, %	5.32 ± 1.13	5.51 ± 1.52	5.59	5.1 ± 0.27	5.01	0.699
hsCRP ^18^, mg/L	10.58 ± 11.75	7.12 ± 2.88	9.33	14.44 ± 16.02	12.22	0.046 *
IMT ^19^, mm	0.88 ± 0.15	0.93 ± 0.18	0.93	0.83 ± 0.1	0.82	0.859

* Mann-Whitney U Test and Chi Square Test significant difference; *n* = absolute number of patients; % = relative number of patients; ^1^ HTN, hypertension; ^2^ SBP, systolic blood pressure; ^3^ DBP, diastolic blood pressure; ^4^ SSS, superior sagittal sinus; ^5^ LS, lateral sinus; ^6^ PS, petrosal sinus; ^7^ SS, sigmoid sinus; ^8^ Cav. S, cavernous sinus; ^9^ Factor V *G1691A*- Leiden gene mutation (polymorphic variant); ^10^ Factor II, prothrombin *G20210A* gene mutation (polymorphic variant); ^11^ PAI-1, plasminogen activator- inhibitor gene mutation (polymorphic variant); ^12^ tHcy, homocysteine; ^13^ TC, total cholesterol; ^14^ LDLc, low- density lipoprotein cholesterol; ^15^ HDLc, high- density lipoprotein cholesterol; ^16^ TGL, triglycerides; ^17^ HbA1c, glycohemoglobin; ^18^ hsCRP, high sensitive C- reactive protein; ^19^ IMT, intima- media thickness.

**Table 2 brainsci-11-00023-t002:** Variable comparisons for *MTHFR* C677T gene polymorphism (*n* = 44) and *MTHFR A1298C* gene polymorphism (*n* = 16) using Mann- Whitney U Test, Chi Square Test and Fisher’s Test.

Variable	With *MTHFR C677T* Polymorphism (*n* = 44)		With *MTHFR A1298C* Polymorphism (*n* = 16)		*p*
	Mean +/− Std Deviation (Median)	Mean Rank	Mean +/− Std Deviation (Median)	Mean Rank	
Age, years	35.3 ± 8.47	35.89	34.36 ± 8.5	33.76	0.548
Male, n (%)	16 (36.37%)	13.76	6 (37.50%)	8.23	*p <* 0.001 *
Female, n (%)	28 (63.63%)	23.95	10 (62.50%)	14.04	*p <* 0.001 *
HTN ^1^, n (%)	10 (22.72%)	8.66	4 (25.00%)	5.33	0.002 *
SPB ^2^, mmHg	127.61 ± 22.51	131.03	129.06 ± 22.62	125.63	0.264
DBP ^3^, mmHg	79.88 ± 13.55	82.80	83.75 ± 13.44	80.82	0.326
SSS ^4^/SSS + LS ^5^	25 (56.81%)	20.88	7 (43.75%)	11.11	*p <* 0.001 *
LS/LS + SP ^6^/LS + SS ^7^, n (%)	15 (34.09%)	12.73	5 (31.25%)	7.26	*p <* 0.010 *
Cav. S ^8^, n (%)	4 (9.09%)	4.09	4 (25.00%)	3.90	0.722^
Factor V ^9^, n (%)	16 (36.36%)	12.71	2 (12.50%)	5.28	*p <* 0.001 *
Factor II ^10^, n (%)	0	0.78	3 (18.75%)	2.21	0.076 ^
PAI-1 ^11^, n (%)	38 (86.36%)	31.83	11 (68.75%)	17.16	*p <* 0.001 *
tHcy ^12^, μmol/L	45.05 ± 34.13	38.82	17.19 ± 34.31	23.41	*p <* 0.001 *
TC ^13^, mg/dL	198.45 ± 38.36	206.16	209.81 ± 38.17	202.09	0.108
LDLc ^14^, mg/dL	115.2 ± 26.74	118.06	115.62 ± 27.51	112.75	0.292
HDLc ^15^, mg/dL	50.36 ± 12.97	52.53	54.06 ± 12.74	51.88	0.385
TGL ^16^, mg/dL	135.81 ± 75.56	141.75	146.12 ± 75.95	140.17	0.149
Blood glucose ^17^	100.31 ± 39.84	103.17	102.12 ± 39.63	99.25	0.316
HbA1c ^18^, %	5.5 ± 1.53	5.64	5.55 ± 1.53	5.40	0.817
hsCRP ^19^, mg/L	7.25 ± 2.93	7.28	6.74 ± 2.85	6.70	0.745
IMT ^20^, mm	0.92 ± 0.18	0.94	0.94 ± 0.18	0.91	0.923

* Mann-Whitney U Test and Chi Square Test significant difference; ^ Fisher’s test significant difference; *n* = absolute number of patients; % = relative number of patients; ^1^ HTN, hypertension; ^2^ SBP, systolic blood pressure; ^3^ DBP, diastolic blood pressure; ^4^ SSS, superior sagittal sinus; ^5^ LS, lateral sinus; ^6^ PS, petrosal sinus; ^7^ SS, sigmoid sinus; ^8^ Cav. S, cavernous sinus; ^9^ Factor V *G1691A*- Leiden gene mutation (polymorphic variant); ^10^ Factor II, prothrombin *G20210A* gene mutation (polymorphic variant); ^11^ PAI-1, plasminogen activator- inhibitor gene mutation (polymorphic variant); ^12^ tHcy, homocysteine; ^13^ TC, total cholesterol; ^14^ LDLc, low- density lipoprotein cholesterol; ^15^ HDLc, high- density lipoprotein cholesterol; ^16^ TGL, triglycerides; ^17^ Blood glucose, mg/dL; ^18^ HbA1c, glycohemoglobin; ^19^ hsCRP, high sensitive C- reactive protein; ^20^ IMT, intima- media thickness.

## Data Availability

3rd Party Data. Restrictions apply to the availability of these data. Data was obtained from Timisoara County Emergency Clinical Hospital and are available from the authors with the permission of Institutional Ethics Committee of Clinical Studies of the Timisoara County Emergency Clinical Hospital.

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
