# Peer review of "MTHFR Gene Polymorphisms and Cardiovascular Risk Factors, Clinical-Imagistic Features and Outcome in Cerebral Venous Sinus Thrombosis"

_brainsci, 2020, doi:10.3390/brainsci11010023_

Round 1

Reviewer 1 Report

The authors of the manuscript have carried out a comprehensive study on cerebral venous sinus thrombosis, including genomic polymorphisms and cardiovascular risk factors, as well as imaging diagnosis and outcomes, which is of interest to clinical and basic science readers. I just have some minor concerns:

  1. As the authors have found, there are significant differences on cholesterol levels  between patients with genetic mutation and those in other risk factor groups, since the average age of patients with genetic polymorphisms are younger, how about the medication like statin treatment history on patients with other etiology factors?
  2. I agree with the authors that the patient number is a limitation of the study, subgroups with different genotypes combination, such  as among MTHFR 677T (CC, CT TT genotypes)  +/- A1298C, or C677T +/- V G1691A, will help to further clarify  the effect of genomic variations on Hcy or cholesterol land CRP levels.
  3. MTHFR SNPs are reported to be associated with Hcy metabolism, it would be more educational if the authors could speculate underlying mechanism links among the homocysteine levels and cholesterol metabolism and inflammation, which contribute to cardiovascular events and complications.

Author Response

Dear Reviewer,

We would like to thank you for your help and support!

Comments and Suggestions for Authors

The authors of the manuscript have carried out a comprehensive study on cerebral venous sinus thrombosis, including genomic polymorphisms and cardiovascular risk factors, as well as imaging diagnosis and outcomes, which is of interest to clinical and basic science readers. I just have some minor concerns:

  • As the authors have found, there are significant differences on cholesterol levels  between patients with genetic mutation and those in other risk factor groups, since the average age of patients with genetic polymorphisms are younger, how about the medication like statin treatment history on patients with other etiology factors?”

Patients in the group with other etiologies of cerebral venous sinus thrombosis have a higher average age than the first group, but are still young patients that have an average age of 40.27 ± 14.33. The patients in cause did not have a personal history of pathology in cardiovascular history and some have received statin treatment before only for high cholesterol - but their number is very small so it was not relevant in the study, not being statistically significant.

  • „I agree with the authors that the patient number is a limitation of the study, subgroups with different genotypes combination, such  as among MTHFR 677T (CC, CT TT genotypes)  +/- A1298C, or C677T +/- V G1691A, will help to further clarify  the effect of genomic variations on Hcy or cholesterol land CRP levels. „

In the study of MTHFR polymorphisms compared to other etiologies, in Table 1, the genotype of the two polymorphisms together was taken into account and in this situation all variables have high values: cholesterol, homocystein and pcr compared to normal and are statistical significant.

In the study of MTHFR polymorphisms C677T compared to A1298C; the statistical significance was present only in the case of homocystein value, being higher at the C677T polymorphism, the other variables analyzed had no statistical significance (table in line 333).

  • „MTHFR SNPs are reported to be associated with Hcy metabolism, it would be more educational if the authors could speculate underlying mechanism links among the homocysteine levels and cholesterol metabolism and inflammation, which contribute to cardiovascular events and complications.”

„Our conclusions of the study are on the same line with the conclusions of other studies from literature, perfomed in various parts of the globe, as follows: the higher levels than normal in plasma Hcy levels translates into hyperhomocysteinemia which represents a risk factor the pathophysiology of cardiovascular and cerebrovascular diseases and also in the recovery of the injuries located in the cerebral area as revealed in studies from literature [27-29]. In a study by Tripathi et al., following the North Indian population, MTHFR C677T polymorphism and higher plasma Hcy levels have also been related with coronary artery diseases [30]. MTHFR polymorphism referring to C677T and A1298C; has been implicated in hypercysteinemia revealing an increased occurrence of ischemic and hemorrhagic stroke in the Chinese population [31, 32]. The data from the study of Kumar et al. has revealed the existence of a correlation between higher Hcy levels and MTHFR C677T polymorphism in subarachnoid hemorrhage patients [33]. Studies from literature [34-37] have emphasiezed the role of MTHFR C677T polymorphism in ischemic stroke. MTHFR C677T polymorphism conducts to substitution of alanine to valine at possition 222 which encodes for a thermolabile enzyme with reduced activity. This decrease in enzyme activity could be accountable for higher Hcy levels. In the study by Kumar et al., MTHFR A1298C polymorphism has also been found to be accountable for the elevated Hcy levels in the Indian population [38].” lines 488-503

Reviewer 2 Report

In this study, authors studied the prevalence of MTHFR gene polymorphisms, factor V G1691A-Leiden mutation, prothrombin G20210A mutation, plasminogen-activator inhibitor PAI-1 675 4G/5G mutation, and hyperhomocysteinemia that correlate with cardiovascular risk factors in patients with CVST. They also evaluated additional causes and risk factors associated with CVST, such as local infections, general infections, obstetric causes (pregnancy, puerperium), head injury, direct injury after ethmoidal puncture for frontal sinusitis. The sample size used in the study (N = 114) is not impressive, but acceptable considering the rare occurrence of CVST, and should be enough for a candidate gene study. However, in the current era of omics studies, candidate gene studies are outdated. Hence, the authors need to provide a stronger hypothesis and literature to back up their research. There are also some major items that require authors’ attention:

  • Authors may consider merging table 1 and table 2.
  • Figures are not referenced in the text.
  • Author should also compare i) MTHFR C677T (n = 44) and other etiology (n= 54), and ii) MTHFR A1298C (n= 16) and other etiology.
  • It is better to use the term polymorphism instead of mutation when referring to single nucleotide polymorphisms, such as MTHFR C677T and MTHFR A1298C
  • For some variable, the number of samples in groups is < 5 (e.g. Factor II, Factor V). Therefore, chi-square test results will not be accurate. Authors should use Fisher’s test as an alternative.
  • Abstract line 24 and introduction line 71, instead of “in a selected patient with CVST”, use “in group of patients with CVST”.
  • In abstract, it is not clear what groups (first group and second group) stand for.
  • Overall, the abstract is very long and detailed, so it may be hard to follow for readers. Authors should consider making the abstract more compact.
  • Introduction line 70, put “and” before hyperhomocysteinemia.
  • Introduction line 73, it should be “the most studied genetic variant”.

Author Response

Dear Reviewer,

We would like to thank you for your help and support!

Comments and Suggestions for Authors

In this study, authors studied the prevalence of MTHFR gene polymorphisms, factor V G1691A-Leiden mutation, prothrombin G20210A mutation, plasminogen-activator inhibitor PAI-1 675 4G/5G mutation, and hyperhomocysteinemia that correlate with cardiovascular risk factors in patients with CVST. They also evaluated additional causes and risk factors associated with CVST, such as local infections, general infections, obstetric causes (pregnancy, puerperium), head injury, direct injury after ethmoidal puncture for frontal sinusitis. The sample size used in the study (N = 114) is not impressive, but acceptable considering the rare occurrence of CVST, and should be enough for a candidate gene study. However, in the current era of omics studies, candidate gene studies are outdated. Hence, the authors need to provide a stronger hypothesis and literature to back up their research. There are also some major items that require authors’ attention: „

  • Authors may consider merging table 1 and table 2.

The modification was made: we merged the tables 1 and 2, line 333.

  • Figures are not referenced in the text.

The modification was made, we put the references of the figures in the text line 272 and line 284.

  • Author should also compare i) MTHFRC677T (n = 44) and other etiology (n= 54), and ii) MTHFRA1298C (n= 16) and other etiology.

We have made the comparisons using the values in the tables 1 and 2 (lines 364-377 and lines 473-479)

By comparing the group of patients with MTHFR C677T gene polymorphism (n=44) with the group patients with other etiology of CVST (n=54) we obtained statistically significant increased values for TC (198.45± 38.36 mg/dL vs.188.81±35.16 mg/dL; p <0.005), LDLc (115.2±26.76 mg/dL vs. 94.25±25.44 mg/dL; p<0.005), tHcy (45.05±34.13umol/L vs. 12.23±4.72umol/L; p<0.005), IMT (0.92±0.18mm vs 0.83±0.1mm; p <0.005). Furthermore, in the group of patients with MTHFR C677T gene polymorphism, the following variables: the number of women and the measured values of cavernous sinus were statistically significant lower compared with the group of patients with other etiology of CVST (28 vs 37; 4 vs.9; p<0.005).

We obtained the following conclusions, by comparing the values obtained by the group patients with MTHFR A1298C gene polymorphism (n=16) with the values obtained by the group of patients with other etiology of CVST (n=54): the results showed statistically significant increased values for IMT (0.94±0.18 mm vs.0.83±0.01mm; p<0.005). Also, the results revealed statistically significant decreased values between the two groups regarding the following variables: male (6 vs 17; p<0.005), female (10 vs.37; p<0.005), SSS/SSS+LS (7 vs. 27: p<0.005), LS/LS+SP/LS+SS (5 vs.18; p<0.005).”

  • It is better to use the term polymorphism instead of mutation when referring to single nucleotide polymorphisms, such as MTHFRC677T and MTHFR A1298C

The modification was made in the whole text.

  • For some variable, the number of samples in groups is < 5 (e.g. Factor II, Factor V). Therefore, chi-square test results will not be accurate. Authors should use Fisher’s test as an alternative.

Fisher’s test was used in table 2 as the reviewer suggested for variables with number less than 5 cases sinsus cavernous and factor II. (lines 353-355)

  • Abstract line 24 and introduction line 71, instead of “in a selected patient with CVST”, use “in group of patients with CVST”.

The modification was made as the reviewer has suggested, line 23 in abstract and line 65 in introduction.

  • In abstract, it is not clear what groups (first group and second group) stand for.

In the abstract the term „first” (group) was used to describe the inherited thrombofilia group and the term „second” (group) was used to describe the other etiology group.

  • Overall, the abstract is very long and detailed, so it may be hard to follow for readers. Authors should consider making the abstract more compact.

The modification was made. We tried to modify the abstract in oder to be compact and more inteligible. The abstract is now between lines 19-48.

  • Introduction line 70, put “and” before hyperhomocysteinemia.

The modification was made in the text, line 65.

  • Introduction line 73, it should be “the most studied genetic variant”

The modification was made in the text, line 68.   

Round 2

Reviewer 2 Report

Authors addressed all the comments and made the necessary edits to improve the manuscript. They've also corrected English language related errors. I do not have any additional comments.